# Relationship between students' attitude towards, and performance in mathematics word problems

**Robert Wakhata**[1]*, **Sudi Balimuttajjo**[2], **Védaste Mutarutinya**[3]

**1** University of Rwanda, College of Education, Rwamagana, Rwanda: African Centre of Excellence for Innovative Teaching and Learning Mathematics and Science (ACEITLMS), Rwamagana, Rwanda, **2** Department of Educational Foundations and Psychology, Mbarara University of Science and Technology, Mbarara, Uganda, **3** University of Rwanda, College of Education, Rwamagana, Rwanda

* rwakhata@gmail.com

## Abstract

The study explored the relationship between students' attitude towards, and performance in mathematics word problems (MWTs), mediated by the active learning heuristic problem solving (ALHPS) approach. Specifically, this study investigated the correlation between students' performance and their attitude towards linear programming word tasks (ATLPWTs). Tools for data collection were: the adapted Attitude towards Mathematics Inventory-Short Form (ATMI-SF), ($\alpha$ = .75) as a multidimensional measurement tool, and linear programming achievement tests (pre-test and post-test). A quantitative approach with a quasi-experimental pre-test, post-test non-equivalent control group study design was adopted. A sample of 608 eleventh-grade Ugandan students (291 male and 317 female) from eight secondary schools (both public and private) participated. Data were analyzed using PROCESS macro (v.4) for SPSS version 26. The results revealed a direct significant positive relationship between students' performance and their ATLPWTs. Thus, students' attitude positively and directly impacted their performance in solving linear programming word problems. The present study contributes to the literature on performance and attitude towards learning mathematics. Overall, the findings carry useful practical implications that can support theoretical and conceptual framework for enhancing students' performance and attitude towards mathematics word problems.

## 1.0 Introduction

### 1.1 The concept of students' attitude towards mathematics

The term attitude is the most indispensable concept in contemporary social psychology and science. It is related to emotional and mental entities that drive an individual towards performing a particular task [1]. According to [2], attitude is "a learned disposition or tendency on the part of an individual to respond positively or negatively to some object, situation, concept or another person" (p. 551). [3], define attitude towards mathematics as positive, negative, or neutral feelings and dispositions. Attitude can be bi-dimensional, (a person's emotions and

**Data Availability Statement:** The data is freely available and accessible to explore and reuse. Repository name: Mendeley. Data identification number: doi:10.17632/xc2xj3rwp9.1. Direct URL to data: https://data.mendeley.com/datasets/

xc2xj3rwp9/1. Supplementary material associated with this article can be found, in the online version, at https://doi.org/10.1016/j.dib.2023.109055.

**Funding:** This research received support from the African Centre of Excellence for Innovative Teaching and Learning Mathematics and Science ([ACEII(P151847)], University of Rwanda, College of Education. The funders had no role in study design, data collection and analysis, decision to publish, or preparation of the manuscript.

**Competing interests:** The authors have declared that no competing interests exist.

beliefs) or multidimensional (affect, behavior, and cognition). Over the last decades, an extensive body of research from different settings and contexts has investigated variables that influence students' attitude towards Science, Technology, Engineering and, Mathematics (STEM) (e.g., [2,4,5]). This means that attitude determines and may be used as a predictor for academic achievement. In this study, we are particularly concerned with an exploration of the effect of the heuristic problem-solving approach on students' attitude towards learning mathematics, and the topic of LP in particular. This is due to significant roles LP play in constructing elementary and advanced models for understanding science, technology and engineering (STE).

Numerous studies on students' attitude towards mathematics which is always translated as liking and disliking of the subject have been published (e.g., [6–9]). To some secondary school students, mathematics appears abstract, difficult to comprehend, boring and viewed with limited relationship or relevance to everyday life experiences. Students start learning the subject well but gradually start disliking some topics or the entire subject. They feel uncomfortable and nervous during learning and examinations. This is partly attributed to students' lack of self-confidence, and motivation during problem-solving. To some students, persevering and studying advanced mathematics has become a nightmare. Indeed, some students do not seem to know the significance of learning mathematics beyond the compulsory level. Students may (may not) relate mathematical concepts beyond the classroom environment if they have a negative attitude towards the subject. This may lead to their failure to positively transfer mathematical knowledge and skills in solving societal problems.

Mathematicians have attempted to research and understand affective variables that significantly influence students' attitude towards mathematics (e.g., [7,10–20]). Some researchers and authors have gone ahead to ask fundamental questions on whether or not students' attitude towards mathematics is a general phenomenon or dependent on some specific variables. To this effect, some empirical findings report students' attitudes towards specific units in mathematics to enhance the learning of that specific content and mathematics generally. (e.g., [6,21–26]).

Rather than investigating students' general attitudes toward mathematics, recent research has also attempted to identify background factors that may provide a basis for understanding students' attitude towards and performance in learning mathematics. Thus, students at different academic levels may have negative or positive attitude towards mathematics due to fundamentally different reasons. Yet, some empirical studies have shown existence of significant relationships between students' attitude and performance in mathematics (e.g., [7,19,27–36]). From the above studies, it appears multiple factors for instance students' demographic characteristics, teachers, parents, and the classroom instructional practices influence students' attitude towards learning mathematics.

This paper presents results from a more specific investigation into the relationship between students' attitude towards, and performance in linear programming mathematics word tasks (Appendix 1 in S1 File). This is because studies concerning attitudes towards and achievement in mathematics have begun to drift from examining general mathematics attitudes to a more differentiated conceptualization of specific students' attitude formations, and in different units (topics). Although different attitudinal scales (e.g., [2,34,37]) were developed to measure different variables influencing students' attitude towards mathematics, this study investigated the influence of some of these constructs on students' performance in linear programming. According to the above authors (and other empirical findings), students' attitude towards mathematics may be the consequence of general and specific latent factors. Thus, attitude towards learning LP word tasks was investigated with specific reference to students' performance, controlling for other variables e.g., learning strategies, students' gender, school type, and school location.

## 1.2 Mathematics word problems

[38] define word problems as "verbal descriptions of problem situations wherein one or more questions have raised the answer to which can be obtained by the application of mathematical operations to numerical data available in the problem statement." The authors categorized word problems based on their inclusion in real-life world scenarios. Thus, mathematics word problems play significant roles in equipping learners with the basic knowledge, skills, and, understanding of problem-solving and mathematical modeling. Some empirical findings (e.g., [39]) show that mathematics word problems link school mathematics to real-life world applications. However, the learning of mathematics word problems and related algebraic concepts is greatly affected by students' cognitive and affective factors [18,40]). Mathematics word problems are an area where the majority of students experience learning obstacles [23,40–47]. By contrast, comprehension of mathematics word problems mainly accounts for students' difficulties. Consequently, this has undermined students' competence, confidence and achievement in word problems and mathematization in general.

Yet, mathematical word problems are intended to help learners to apply mathematics beyond the classroom in solving real-life-world problems. [38,39] have argued that mathematics word problems are difficult, complex, and pause comprehension challenges to most learners. This is because word problems require learners to understand and adequately apply previously learned basic algebraic concepts, principles, rules and/or techniques. Indeed, most learners find it difficult to understand the text in the word problems before transformation into models. This is partly due to variation in their comprehension abilities and language [48]. Consequently, learners fail to write required mathematical algebraic symbolic operations and models. Yet, incorrect models lead to wrong algebraic manipulations and consequently wrong graphical representations and solutions.

Notably, research findings by [12] indicate that students' mathematical proficiency is partly explained by their transitional epistemological and ontological challenges from primary to secondary education. This may consequently affect their attitude towards mathematics during the transition from secondary to tertiary [11]. The authors attribute this trend to mainly three dimensions (emotional disposition, vision of mathematics and perceived competence). Other studies (e.g., [28,49–51] attribute students' performance and achievement in mathematics to gender differences. Research by [52] shows that the challenges experienced in mathematics education are a by-product of those in education in general, and they span from policy, curriculum, instruction, learning, and information technology to infrastructure. Thus, students may start excellently learning and performing mathematics from primary but gradually lose interest in some specific units and finally in mathematics generally. Several strategies have been adopted and/or adapted to boost students' attitude in specific topics. For the case of LP, students' attitude towards mathematics and in equations and inequalities in particular gradually drop in favor of other presumably simpler topics. To boost performance in mathematics word problems, [44] proposed step-by-step problem-solving strategies to enhance mastery and performance.

Students' attitudes should, therefore, be investigated as well as their influence on their conceptual changes. Several empirical studies have also investigated the relationship between attitude towards, and achievement in mathematics across all levels, and in different contexts (e.g., [7,9,20,24,53–61]). In particular, these studies generally focused on students' attitude towards mathematics, and many of them from the western context [62]. Yet, students may have different perceptions and attitudes towards specific content (topics) in mathematics irrespective of their setting, context and the learning environment.

Several authors (e.g., [63–68]) have highlighted numerous difficulties encountered by students in solving algebraic inequalities. These difficulties include transformation of

mathematics word text into models, wrong graphical representations, wrong optimal solutions; etc. Thus, a combination of methods (strategies) rather than one specific method [69] (e.g., heuristics, problem solving, group discussions, giving frequent exercises, etc.) can be applied to minimize specific students' learning challenges. To enhance mathematical conceptual proficiency, educators should boost students' cognitive and affective domains in specific mathematics content. This is because students' affective domain may directly influence their cognitive and psychomotor domains. In this study, it was predicted that students' proficiency in solving LP word tasks is largely influenced by their attitude and inadequate prior algebraic knowledge, skills, and experiences. [23] noted that prior conceptual understanding coupled with students' attitudes towards solving algebraic concepts impacted students' inherent procedures in writing relational symbolic mathematical models from word problems, and provision of correct numerical solutions.

## 1.3 Attitude towards mathematics word problems

Improving students' achievement and attitude in mathematics is the recent global talk and practice in the 21st century education systems. Educational stakeholders seem to agree in unison that learning experiences, achievement and success in mathematics is a function of attitude [70]. Researchers have argued that learners' positive attitude toward mathematics is key. Fostering a positive attitude towards mathematics means making the learning of different topics, related mathematical concepts and experiences positive. Whether it is classroom learning activities, homework, practice, or a test, it means supporting and encouraging learners to perform the assigned tasks so that they are motivated, feel confident about their mathematical skills, and during problem-solving. This is because, learning mathematics goes beyond memorization and passing examinations in a school setting. It is about going an extra mile to solve problems, and this helps greatly in applying mathematics outside the classroom environment in real-life scenarios. Thus, as a child grows and steps into the real world, mathematics helps a great deal in achieving the skill of decision making and problem-solving.

In relation to achievement, a positive attitude towards mathematics increases learner's odds to select mathematics courses in high school and beyond, especially in making mathematics related career choices. A number of researchers endeavored to formulate a valid and reliable measure of mathematics attitude. Some of the attitudinal scales were measured uni-dimensionally while others multidimensionally (e.g., [2,33] developed scales designed to measure enjoyment and value of mathematics. This led to Fennema-Sherman Attitudinal scales. According to [34], this tool has been adapted to measure students' attitude in mathematics education research in different contexts and settings for the last three decades. Specifically, the Fennema-Sherman Mathematics Attitude Scales consist of a group of nine instruments with 108 items. These subscales include (1) attitude toward success in mathematics scale, (2) mathematics as a male domain scale, (3) and (4) mother/father scale, (5) teacher scale, (6) confidence in learning mathematics scale, (7) anxiety in mathematics scale, (8) motivation in mathematics scale, and (9) value (usefulness) in mathematics scale.

The attitude towards mathematics inventory (ATMI) measures attitude in terms of four sub-scales (self-confidence, enjoyment, motivation, and value). Self-confidence sub-scale assesses the level of confidence to do mathematics; enjoyment taps the extent to which a student enjoys doing mathematics; motivation refers to the level of motivation to do mathematics; and value assesses the extent to which the student attaches value to doing mathematics. The ATMI has a strong construct validity and reliability [71]. Moreover, recent findings (e.g., & [72,73]) have revealed a strong positive relationship between students' attitude towards, and achievement in mathematics.

## 1.4 Linear programming

The history of linear programming is dated as far back as in the 1940s. The concept of linear programming arose due to the need to find solutions to complex planning problems during World wartime operations [74]. Linear programming was used extensively during World War II to deal with the transportation, scheduling, and allocation of resources subject to certain restrictions (e.g., costs and availability). Today, linear programming is still being applied in business and industry, specifically in production planning, transportation and routing, and in various types of scheduling. In addition, and to date, airlines use linear programs to schedule their flights, taking into account both scheduling aircraft and scheduling staff. Developed by [74], the simplex method was efficiently applied to solve linear programming problems. Later, the Neumann, also established the theory of duality to complement the simplex method. Presently, the above methods of solving linear programming problems are still being taught to students at university and other tertiary institutions.

Linear programming is also one of the topics in Ugandan secondary school mathematics curriculum. It is one of the topics that require students' understanding of basic mathematical algebraic principles and rules. It is a basic introduction to other advanced methods for solving and optimizing LP problems. At secondary school level, solving LP problems is a classical unit, "the cousin" of mathematics word problems which has gained significant applications in the last decades in mathematics, science, and technology [75–79]. This is because LP links theoretical to practical mathematical applications. The topic provides elementary modeling skills for later applications in modelling.

Previous empirical studies have revealed that LP and/or related concepts are not only difficult for learners but also challenging to teach [40–42,44,80]. Different cognitive factors account for learners' challenges in mathematics word problems [81,82]. The learning challenges mainly stem from students' difficulties in comprehending mathematics word problem statements, application of suitable algebraic principles, negative attitude, and transformation from conceptual to procedural knowledge and understanding [18,38,40–42]. Learners' attitudes toward solving mathematics word problems should, therefore, be investigated and integrated during classroom instruction to help educational stakeholders provide appropriate and/or specific instructional strategies.

## 1.5 The Ugandan context

In Uganda, studies on predictors of students' attitude towards science and mathematics are scanty. There seems to be no recent empirical findings on students' attitude or performance in mathematics and mathematics word problems in particular. Solving LP tasks (by graphical method) is one of the topics taught to the 11th grade Ugandan lower secondary school students [83,84]. Despite students' general and specific learning challenges in mathematics, the objectives of learning LP are embedded within the aims of the Ugandan lower secondary school mathematics curriculum. Some of the specific aims of learning mathematics in Ugandan secondary schools include. . .enabling individuals to apply acquired skills and knowledge in solving community problems, instilling a positive attitude towards productive work . . ." [84]. Generally, the learning of LP word problems aims to develop students' problem-solving abilities, application of prior algebraic conceptual knowledge and understanding of linear equations and inequalities in writing models from word problems, and from real-life-world problems. Despite the learning challenges, the topic of LP is aimed at equipping learners with adequate knowledge and skills for doing advanced mathematics courses beyond the compulsory level at Uganda Certificate of Education (UCE).

However, every academic year, the Uganda National Examinations Board (UNEB) highlights students' strengths and weaknesses in previous examinations at UCE. The consistent reports [85–88] on previous examinations on work of candidates show that students' performance in mathematics is not satisfactory especially at distinction level. In particular, previous examiners' reports show students' poor performance in mathematics word problems. The examination reports further revealed numerous students' specific deficiencies in the topic of LP (Appendix 1 in S1 File). Students' challenges in LP mainly stem from comprehension of word problems to formation of wrong linear equations and inequalities (in two dimensions) from the given word problem in real-life situations. Thus, wrong models derived from questions may result in incorrect graphical representations, and consequently wrong solutions and interpretations of optimal solutions. These challenges (and others) may consequently hinder and/or interfere with students' construction of relevant models in science, mathematics and technology. Moreover, learners have consistently demonstrated cognitive obstacles in answering questions on LP, while others elude these questions (Appendix 1 in S1 File) during national examinations. Noticeably absent in all the UNEB reports are factors that account for students' weaknesses in learning LP and the specific interventions to overcome students' challenges in LP. Some students have developed a negative attitude towards the topic. Yet, students' attitudes may directly impact on their learning outcomes [37].

Several attitudinal scales (with both cognitive and behavioral components) have been developed [61,89] adopted or adapted [3] to assess students' attitude towards mathematics and in specific mathematics content. For instance, Geometry Attitude Scales [90], Statistics Attitude Scales [57,91], Attitudes toward Mathematics Word Problem Inventory [40], the Attitude Towards Geometry Inventory (ATGI) instrument [92], and others. In this study, we adapted the Attitude towards Mathematics Inventory-Short Form (ATMI-SF) instrument [3] to investigate the effect of the heuristic problem-solving approach on students' attitude towards learning LP word problems. Specifically, this research explored the 11th-grade students' attitude towards learning LP word problems (see Appendix 1 in S1 File). Taken together, research shows that a high percentage of educational stakeholders around the world are concerned about performance and attitude towards mathematics word tasks in particular. However, to fully understand students' attitude towards and performance in mathematics, it is necessary to investigate beyond general mathematics attitudes and examine specific underlying aspects for these attitudes.

## 1.6 Objectives of the study

The purpose of this study was to explore the relationship between students' attitude towards, and performance in LP mathematics word problems. Specifically, the present study aimed at:

-Investigating the direct effect of students' attitude towards, and performance in mathematics word problems.

-Exploring the indirect effect of students' attitude towards, and performance in LP word tasks mediated by active learning heuristic problem solving strategies.

## 1.7 The Theoretical framework

This study was situated on the theoretical framework according to constructivism, and Eccles, Wigfield, and colleagues' expectancy-value model of achievement motivation [93]. The expectancy-value model is based on the expectancy-value theories of achievement. The theory combines the motivational components of competency beliefs, importance of the subject, and utilitarian beliefs, and focuses on both the role of students' beliefs about their competence and

the value they place on the activity. The Expectancy-Value Theory is a theory of motivation that describes the relationship between a student's expectancy for success at a task or the achievement of a goal about the value of task completion or goal attainment.

Expectancy refers to a student's expectation for success on a given task (in this case, LP tasks). In addition, the social constructivist theory guided the epistemological and pedagogical perspectives. This theory is grounded in the notion that learners construct their knowledge and understanding through participatory learning and interaction with one another within the prevailing learning environment. The learners' prior basic cognitive knowledge and experiences influence subsequent learning practices, as witnessed through assessment procedures [68]. The constructivist theory is based on the premise that success on specific tasks and the values inherent in those tasks is positively correlated with students' previous experiences, achievement, and consequently students' prior abilities. Constructivism is directly linked to the Expectancy-Value Theory, students' levels of cognitive demand, and the teachers' pedagogical content knowledge [94].

This research is a subset of a large study that investigated the effect of active learning through the heuristic problem-solving approach on students' achievement and attitude towards learning mathematics word problems and LP in particular. It is expected that the findings will be applied to enhance mastery of students' mathematical concepts within the learners' zone of proximal development. This can be attained by engaging learners, devising effective procedures, and application of prior conceptual and procedural knowledge in subsequent learning. In so doing, educators and learners may devise suitable and multiple problem-solving methods and develop a positive attitude towards learning. To achieve the objectives of this study, the framework provided by [94] was used to map students' LP cognitive demands. Cognitive demand was described by [94] as "the kind and level of thinking required of students to successfully engage with and solve the task" ([94] p.11).

[94] interpreted these levels as problem-solving strategies. According to [94] teachers should take into account different levels of cognitive demand with varying mathematics tasks given to students. The authors reasoned that students' mathematical proficiency and competency are determined by the tasks they are given during instruction. Mathematics tasks at the lower cognitive stage (memorization level, and connection with procedures), for example, must be different from those at the highest cognitive level (connection without procedures, and doing mathematics). During task review, this approach supports teachers' instructional activities and approaches (what is learned, how it is learned, and when it should be learned) following Stein et al.'s levels of cognitive demand. In the context of learning LP, students should first understand and appropriately apply the previous knowledge of equations and inequalities in solving LP non-routine mathematics word problems.

The practical aspect of [94] lies in its implementation [94]. Indeed, if students are only exposed to solving memorization tasks, they may not be able to adequately master non-routine high-level (higher-order thinking) tasks that require critical thinking skills (doing mathematics). Thus, as students advance through their academic stages, teachers need to adjust and involve them in answering high-level tasks from the beginning to improve their critical thinking, and problem-solving abilities. To effectively achieve this, [94] suggested four levels of cognitive demand: two lower-level demand tasks (memorization and procedures with connection to concepts) and two higher-level demand tasks (procedures without connections and doing mathematics). According to Stein et al., students' proficiency in "doing mathematics" tasks may improve their problem-solving abilities, especially in solving non-routine tasks in new contexts. In this case, cognitive level characteristics formed a conceptual framework for evaluating individual students' levels of cognitive demand in learning LP tasks. These characteristics are important in the sense that they highlight specific cognitive levels needed for students to

correctly perform LP mathematics word tasks as well as specifying students' cognitive levels at the time of administering LP tasks (pre-test and post-test).

To link the above two theories (Expectancy-Value Theory and social constructivism), this research also applied the Pedagogical Content Knowledge (PCK) conceptual framework based on [95]). conceptualized that "pedagogical content knowledge also includes an understanding of what makes the learning of specific topics easy or difficult: the conceptions and preconceptions that students of different ages and backgrounds bring with them to the learning of those most frequently taught topics and lessons" (p. 9). According to [95], effective learning strategies involve teachers' integration of students' preconceptions and misconceptions held previously and how these preconceptions relate to subsequent learning. In this case, teachers were expected to apply suitable learning strategies to boost students' understanding of LP concepts. In supporting students' problem-solving strategies, mathematical thinking and understanding, [96] developed a framework for examining mathematics teachers' PKC. The framework is related to Shulman's conceptualization of PCK and is important in enhancing teachers' PCK (e.g., the use of graphics, and manipulatives) with the main objective of understanding students' learning challenges, their mathematical thinking and reasoning.

## 1.8 Conceptual framework of the study

Based on the theoretical framework discussed above, a conceptual framework shows the direct relationship between students' attitude towards, and performance in mathematics word problems, mediated by active learning heuristic problem solving strategies. Active learning encompasses different instructional strategies that promote students' engagement and active participation in constructing knowledge and understanding of particular concepts. These strategies may take the form of hands-on activities, problem-solving tasks, critical thinking etc. This approach involves learners' individual or collaborative performance on routine and non-routine tasks. In this case, learners are directly involved in thinking and doing through discussions, reviews, evaluations, concept maps, role plays, hands-on projects, and cooperative group studies. Often, active learning tasks require learners to make their thinking explicit, allowing educators to gauge and understand students' learning.

One major benefit of active learning is that it develops students' higher-order thinking skills (analysis, synthesis, and evaluation) and also prepares them to apply mathematics in real-life scenarios. Several studies have shown that students in typical active learning classrooms perform better than those taught conventionally [97,98]. This is because they have an opportunity to reflect, conjecture, or predict outcomes, and then to share and discuss their concepts with teachers and their peers to activate and re-activate their cognitive processes. Active learning helps students to reflect on their understanding by encouraging them to make connections between prior mathematical knowledge and new concepts. The conceptual model hypothesizes the existence of a direct relationship among the stated variables. In particular, the conceptual model is shown in Fig 1 below. The model assumes the existence of a relationship between attitude and performance in post-test (including pre-attitude and post-attitude). There exist other indirect inherent relationships which are not part of this study.

## 2.0 Materials and methods

A quantitative survey research approach [99–101], was used to collect, analyze, and describe students' experiences and latent behavior in learning linear programming mathematics word tasks. The present study was guided by a philosophical pragmatic paradigm. Pragmatists believe that the world has several realities, and that there is no unique way of describing and interpreting these realities. The pragmatism paradigm aligns with the quantitative approach

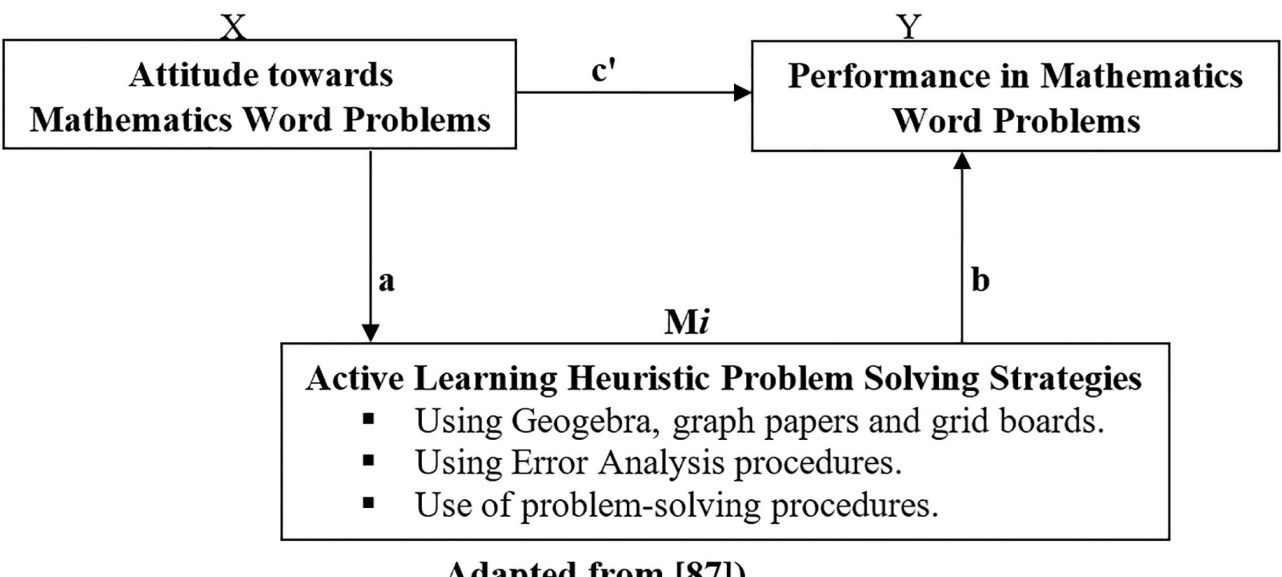

**Fig 1. Showing direct (c') and indirect (b.c) relationship between attitude towards and performance in mathematics word problems mediated by active learning heuristic problem solving strategies.**

[102,103]. This paradigm was suitable for studying and understanding the underlying relationships between the application of active learning through the heuristic approach (independent variable) and students' achievement and attitude towards learning LP (dependent variable).

## 2.1 Research design

This study investigated the relationship between students' attitude towards, and performance in mathematics word tasks. The study adopted a quantitative approach to gain a deeper and broader understanding [99,102,104]. A quasi-experimental pre-test, post-test, non-equivalent control group research design was adopted with the objective of observing the response of subjects to the treatment [105]. The quasi-experimental design was applied to observe and examine individual students' attitude, and achievements in LP, including their conceptual and procedural understanding of LP concepts. It was also used to establish how and why, and to provide similar findings like in cases where a typical classical (true) experimental design is applied [105]. The researchers' awareness of non-random selection could influence the possibility of errors in the final results [106]. However, the justification is that the more similar intact classes for experimental and comparison groups are in terms of baseline characteristics, the closer the results of the quasi-experimental non-equivalent control group research design approximate to the results of a true experimental design. By using the stated approach and design, researchers ably compared and contrasted students' attitude towards learning LP word problems. Learners from the sampled schools (experimental and comparison groups), and in their intact classes participated. Intact classes were maintained to avoid interfering with the respective schools' set timetables and schedules.

## 2.2 Sample and sampling methods

The population includes all people with related characteristics who qualify to be respondents in a given study [106] while the target population is the specific group of respondents on

whom a researcher plans to generalize the sample research findings [107]. A sample is a group of respondents selected from the target population [102]. It is laborious, expensive and time consuming to conduct a census survey. Most often, the population is large and scattered geographically. Thus, sampling techniques become inevitable. In this study, the sampling frame was mutually homogeneous and internally heterogeneous. The sampling frame (units of analysis) were all the 11th grade students for the academic year 2020/2021. Both private and public secondary schools (rural and urban) participated.

Thus, the sampling frame was clustered using a multi-stage cluster sampling method, followed by purposive and convenient sampling [102,107]. Clusters were selected using simple random sampling. The first stage involved the selection of two districts stratified by regions (Mbale district in the Eastern region and Mukono district in the Central region respectively). The multistage cluster sampling method was used to select four secondary schools from each region. First, the two regions were selected based on student's previous academic performance at UCE and inherent school characteristics. Second, purposive and convenient sampling were used to select four secondary schools from each region (two schools from each region were assigned to the experimental group). To avoid sampling errors [105], the use of purposive and convenient sampling aided the selection of schools within the study context with similar (almost similar) educational characteristics (government/private, rural/urban, with low/high enrolment) for experimental and comparison groups.

In this study, researchers carefully selected subjects with a purpose in mind, and based on the believe that the selected subjects had the key information for the study [105,106]. One or more specific predetermined groups were considered as likely to have and capable of providing the required data. The purpose of purposive sampling was to permit in-depth exploration subject to the available resources, time and the theoretical saturation [108]. It was intended to access the target sample as quickly as possible especially where sampling for proportionality was not the primary concern. It was also intended to obtain the opinions of the target population easily.

Finally, to avoid disrupting the already existing research setting in sampled secondary schools, all students in their intact classes constituted the units of analysis. According to [108], multi-stage cluster sampling is suitable due to its flexibility, saves time and money, and is applicable in the collection of primary data from a heterogeneously dispersed population. Selected secondary schools for the experimental group were approximately 250 km away from one another for two reasons; first, to avoid diffusion and spurious results, and second, to compare and contrast students' abilities, achievements and their attitude towards LP word tasks in the two regions. The units of analysis in their intact classes were all selected as clusters to represent the entire sampling frame. A sufficient number of respondents from the entire population was selected in such a way that the sample findings would be used to make generalizations to the whole population [106].

## 2.3 Participants

To compensate for non-response and considering intact classes, 608 respondents were selected from a heterogeneously large population to achieve precision and external validity. There were 639 students at the time of administering the pre-test achievement tests and questionnaires. During data entry and analysis, the scores of thirty-one (31) students from the experimental group were excluded due to missing post-test average scores. The students missed the post-test because they were absent on the day the post-test was administered. Consequently, the findings reported in this research were based on data from 608 students from eight randomly selected secondary schools. Hence, the sample size (n) = 608 [109]) was adequate. According

to [107], the application of [109] is appropriate when the preferred margin of error is 0.05, which is in confirmation with the above sample size.

Four schools were selected from Mbale district, eastern Uganda, and four from Mukono district, central Uganda. Schools were allocated to the experimental and comparison groups by a toss of a coin. Two schools from each region were assigned to the experimental group. Selection of the 11 grade students was based on the content covered in this class as outlined in the Ugandan mathematics curriculum materials [84]. Of the 608 students, 291 (47.86%) were males and 317 (52.14%) were females with a mean age of 18.36 (S.D = 0.94) years. Three hundred seventeen (52.1%) students were from the comparison group while two hundred ninety-one (47.9%) were from the experimental group. The selection of students from the two distant schools within/outside the regions and assigning them to experimental group was to avoid spurious results. A situation where a particular school had more than one class "stream", at least one hundred students with varied academic abilities were randomly selected from different classes to respond to the attitudinal questionnaires. The school (subject) heads revealed that the mathematics syllabus containing LP word problems (Appendix 1 in S1 File) had been completed at the time of data collection. The students were selected to provide their experiences and attitudes toward learning LP word problems.

## 2.4 Ethical considerations

Ethical clearance was sought from the Research and Ethics committee of corresponding authors' university (Directorate of Research and Innovation, University of Rwanda–Ref: 03/DRI-CE/061(b)/EN/gi/2020). Subsequent permission was sought and granted by the permanent secretary ministry of education and sports, the district education officers, and finally from the headteachers of sampled secondary schools before accessing research sites. Upon accessing research participants, they were informed and clearly explained to the purpose of the study. Written informed consent was obtained from participants of at least 18 years. Where necessary, consent from parents or guardians was obtained for respondents below 18 years. Finally, identification numbers were allotted to participants before they anonymously and voluntarily completed questionnaires. Questionnaires were administered to the respondents during school working hours without interfering with the school set timetables. The heads of the sampled schools provided appropriate schedules and personnel to help the principal researcher together with research assistants to quickly and effectively administer questionnaires. All participants were assured of confidentiality and anonymity before they willingly consented. Those who opted not to participate even after the distribution of questionnaires were allowed to withdraw.

## 2.5 Research instruments and procedure for administration

In addition to demographic questions, the Attitude Towards Mathematics Inventory-Short Form (ATMI-SF) [3], a 14-item instrument questionnaire consisting of four subscales (enjoyment, motivation, value/usefulness and self-confidence) was adapted (Appendix 2 in S1 File) to measure the relationship between students' performance and their attitude towards learning LP word tasks. The ATMI-SF, a 5-point likert-type scale with response options ranging from "Strongly Disagree (1)" to "Strongly Agree (5)" was used. The ATMI-SF items were developed from [89], which were also developed and validated from several mathematics attitudinal questionnaire items [2,31,34,62,110]. The ATMI-SF was adapted (Appendix 2 in S1 File) because it directly correlates with the learning of LP, "the cousin of mathematics word problems." English being the language of instruction in Ugandan secondary schools' curricula, translation of questionnaire items was not required.

Content validity of the questionnaire was assessed by three experts (one senior teacher for mathematics, one senior lecturer for mathematics education, and one tutor at teacher training institution). The experts were selected based on their vast experience in teaching mathematics at various academic levels. They evaluated the appropriateness and relevance of the adapted questionnaire items. Based on their recommendations, suggestions and comments, some questionnaire items were adjusted to suit students' academic level and language to adequately answer the research objectives. Reliability of the construct was ascertained using Cronbach alpha (0.75). This threshold is acceptable based on recommendations from [111].

The ATMI-SF was administered to the experimental and comparison groups. Both the experimental and comparison groups were taught LP (equations, inequalities and LP) following the designed and approved Ugandan mathematics curriculum materials [84]. The experimental group was taught LP using the heuristic problem-solving approach before and after an intervention. The learning in the comparison group was purely conventional, and teachers did not follow proper and organized strategies as it was the case with the experimental group. In particular, students from the experimental group were taught LP using several active learning heuristic strategies following clearly outlined principles and strategies.

To adequately implement active learning heuristic problem-solving strategies, teachers from the experimental group were trained. First, students' prior conceptual knowledge of equations and inequalities plus the basic algebraic principles and understanding were reviewed to link previous concepts to learning of LP. Second, several learning materials were applied to help students adequately master LP concepts. The materials included the use of graphs, grid boards, excel and GeoGebra software (see Appendix 3 in S1 File for G01, G02, and G03). Empirical studies (e.g., [42,112,113]) have found that the stated materials enhance students' conceptual understanding and critical thinking abilities. These strategies were further integrated in problem solving strategies [114] by ensuring that students understand the LP word problem, devise a plan, adequately carry out the plan and finally look back to verify solution sketches (see Appendix 3 in S1 File for PS01, PS02, PS03, PS04). To ensure that students minimize errors and misconceptions, the learning of LP was further integrated with Newman Error Analysis (NEA) model [115]).

The instrument was adapted and validated by mathematics education experts as an error analysis tool to adequately provide teachers with a framework to examine underlying reasons why students make errors and misconceptions when answering mathematics word problems. In this tool, teachers emphasized question reading and decoding, comprehension, transformation, process skills and encoding (see Appendix 3 in S1 File for NEA01, NEA02, NEA03, NEA04 and NEA05). Effective utilization of this tool means that students are capable of understanding, confidently and correctly answering mathematics word problems. Previous errors and misconceptions can easily be identified, and applied in subsequent learning to boost students' conceptual and procedural understanding hence providing independent reasoning to their own solutions. Error analysis also helps educators to identify factual, procedural, and conceptual mistakes commonly made by students to provide academic and conceptual support whenever it is needed. In conclusion, this research will provide remedies for correcting students' challenges in learning mathematics word problems, and to boost their problem solving and critical thinking skills.

The mathematics word problem-solving strategies supported students' thinking, and reasoning. Consequently, teachers' identified students' preconceptions, misconceptions and errors as they solved LP word tasks. Multiple representations and demonstrations were carried out to help students understand preliminary concepts and use them to optimize LP tasks. Students were engaged in the typical PS scenario individually, in pairs and in small groups. Questions and specific classroom tasks ranged from simple to complex and from concrete to

abstract. Materials for instance graph papers were provided as teachers guided and demonstrated the graphical solution of LP problems. Teachers provided procedures by supporting and guiding learners during the learning process. All students' preconceptions, misconceptions and errors were addressed. For instance, confusion on whether or not to use dotted lines or solid lines to represent the inequality $y>$a+b$x$, plotting lines to represent an equation or inequality, shading wanted or unwanted regions, etc.

All learners were assigned varied tasks to apply suitable strategies for yielding the optimal solution of a LP word problem. The "bright" students were allowed to share their experiences with the low attainers in pairs or in their small groups. Where necessary, local language was allowed for students' peer instruction in pairs or small groups to master the underlying concepts. The classroom interaction and presentations enhanced students' conceptual knowledge and procedural understanding of LP concepts. This further improved their reasoning, creativity and critical thinking. Consequently, constructive and informative feedback to retain their conceptual understanding was fostered. All challenging concepts on LP were harmonized by individual teachers during reflection and the evaluation phase. All students who had challenges even after this phase were allowed to repeat the tasks by doing corrections and submitting their work for re-marking. The intervention took approximately three months from October 2020 to February 2021. To find out whether or not active learning heuristic problem-solving approach changed students' attitude and performance, the post attitude questionnaire and achievement test were administered to both groups by the principal researcher (assisted by research assistants), and both descriptive and inferential analysis was done using SPSS (V.26). Students' feedback on attitudinal constructs and the post-test scores were compared and contrasted. Finally, this study established if there was a significant relationship between students' performance and their attitude towards mathematics word tasks, mediated by active learning heuristic problem-solving approach (Appendix 3 in S1 File).

## 4.0 Results

### 4.1 Procedure for data analysis

The ATMI-SF questionnaires were completed by the sampled students at their respective schools in their natural classroom setting. The questionnaires were completed in at most 30 minutes on average. The survey instrument contained a 'filter statement', as a Social Desirability Response (SDR) to verify and discard respondents' questionnaires especially those who did not read (see item 15 in Appendix 1 in S1 File) or finish answering all questionnaire items [116,117]. Written consent was received from all participants and participation in this study was completely voluntary and confidential. Participants who felt uncomfortable to complete the questionnaire items were not penalized. Data were collected with the help of mathematics heads of department who were selected from sampled schools on the basis of their expertise and experience. Participants were explained to, the purpose of the study before administering and/or filling in of questionnaire items. In the presence of the principal researcher, research assistants and some selected school administrators, participants completed and returned all the questionnaires. Descriptive and inferential statistics were used to analyze the collected data with reference to the background characteristics and stated hypothesis. Data were analyzed using the Statistical Package for Social Sciences (SPSS) version 26, with [118] PROCESS (v.4) macro. This provided the analysis for exploring whether or not there exist a significant relationship between students' performance and their attitude towards LP mathematics word problems.

## 4.2 Findings and interpretation

**Psychometric properties.** IBM SPSS (version 26) software package was used for analysis. Preliminary statistical analysis revealed no evidence of missing data due to a few cases which were ignored because they did not exceed 5% of sample cases [89,119,120]. However, out of 639 questionnaires distributed, 31 questionnaires were removed because the participants did not either conform to SDR [116,117] or the questionnaires were incomplete. Univariate analysis was run to examine the degree of normality [111,121]. The indices for skewness and kurtosis were within the acceptable ranges (±2 and ±7 respectively) [111,122]. Thus, data were fairly normally distributed.

We tested the psychometric properties (reliability and factor analysis) of the two instruments. The Cronbach alpha coefficient of the adapted ATMI-SF was α = 0.75. Factor analysis was performed using the principal component (with varimax rotation) [119,121,123]. The values obtained were consistent with [3,40] findings. The Kaiser-Meyer-Olkin Measure of Sampling Adequacy Test (KMO) and Bartlett's test of sphericity was conducted. The value of KMO = 0.77> 0.60; and that of Bartlett's test of sphericity was significant (894.349, p<0.05) indicating a substantial correlation in the data and an acceptable fit. For a self-developed standardized active learning heuristic problem-solving (ALHPS) tool, α = 0.71, KMO = 0.74 > 0.60; and that of Bartlett's test of sphericity was significant (253.092, p<0.05). Following the above recommendations, all items were found to be acceptable with adequate construct validity, internal consistency and homogeneity [121,124]. Overall, these items were deemed fit to measure the relationship between active learning heuristic problem-solving and students' attitude towards linear programming word tasks.

Tables 1 and 2 show descriptive statistics (mean, standard deviation, skewness and kurtosis). The values in the table show students' scores on ATMI-SF and ATLPWTs. The results do not show any significant differences between the relationship between ALHPS approach and students' attitude towards linear programming (enjoyment, motivation, usefulness and self-confidence). Indeed, both experimental and comparison groups were similar during pre-test. There was however a change in students' ATLPWTs due to the intervention administered. The findings show that students generally held negative attitude towards learning LP word tasks. These findings are consistent with other empirical research findings (e.g., [40]).

**Table 1. Descriptive statistics: Students' ATLPWTs by item (see Appendix 2 in S1 File).**

| Items | N | Mean | S.D | Skewness | Kurtosis |
|---|---|---|---|---|---|
| A1 | 608 | 3.16 | .971 | .047 | .089 |
| A2 | 608 | 3.31 | 1.196 | -.237 | -.793 |
| A3 | 608 | 3.43 | 1.064 | -.457 | -.242 |
| A4 | 608 | 3.42 | 1.131 | -.362 | -.536 |
| A5 | 608 | 3.27 | 1.103 | -.360 | -.362 |
| A6 | 608 | 3.43 | 1.073 | -.470 | -.273 |
| A7 | 608 | 3.47 | .941 | -.582 | -.002 |
| A8 | 608 | 3.36 | 1.018 | -.463 | -.333 |
| A9 | 608 | 3.33 | .980 | -.479 | -.203 |
| A10 | 608 | 3.36 | 1.046 | -.302 | -.512 |
| A11 | 608 | 3.26 | 1.028 | -.051 | -.453 |
| A12 | 608 | 3.31 | 1.050 | -.242 | -.385 |
| A13 | 608 | 3.12 | .946 | .060 | -.302 |
| A14 | 608 | 3.29 | 1.111 | -.359 | -.577 |

**Table 2. Descriptive statistics: Students' ALHPS approach (see Appendix 3 in S1 File).**

| Item | N | Mean | S. D | Skewness | Kurtosis |
|------|-----|------|------|----------|----------|
| G01 | 608 | 3.82 | .999 | -.884 | .710 |
| G02 | 608 | 3.75 | .852 | -.451 | -.012 |
| G03 | 608 | 3.51 | 1.071 | -.482 | -.709 |
| PS01 | 608 | 2.49 | 1.205 | .776 | -.302 |
| PS02 | 608 | 2.14 | 1.013 | 1.316 | 1.881 |
| PS03 | 608 | 2.26 | .984 | 1.103 | 1.520 |
| PS04 | 608 | 2.24 | .838 | 1.354 | 3.078 |
| NEA01 | 608 | 2.28 | .919 | 1.266 | 2.225 |
| NEA02 | 608 | 2.32 | .990 | 1.164 | 1.559 |
| NEA03 | 608 | 2.42 | .912 | .846 | 1.375 |
| NEA04 | 608 | 2.31 | .999 | 1.026 | 1.238 |
| NEA05 | 608 | 2.50 | .989 | .911 | .979 |

**Table 3. Model summary for active learning.**

| R | R-sq | MSE | F | df1 | df2 | p |
|-----|------|------|--------|-------|---------|------|
| 313 | .098 | .210 | 65.795 | 1.000 | 606.000 | .000 |

**Table 4. Model for attitude towards mathematics word problems.**

| | Coeff. | S.E | t | p | LLCI | ULCI |
|----------|--------|------|--------|------|-------|-------|
| Constant | 1.613 | .132 | 12.254 | .000 | 1.354 | 1.871 |
| Attitude | .318 | .039 | 8.111 | .000 | .241 | .395 |

Standardized coefficients .3130.

Tables 3 and 4 show that although the data is explained by 10% of the model, the active learning heuristic problem solving approach has a significant effect on attitude ($p < 0.05$). Moreover, the narrow confidence interval (0.154) shows a greater degree of precision.

Tables 5 and 6 show that students' attitude towards mathematics word problems has a direct and significant effect on performance. This model accounted for by approximately 50% This means that a positive change in students' attitude directly affects students' performance and vice versa.

Table 7 shows the significant direct effect of students' attitude towards and performance in LP mathematics word tasks.

However, from Tables 8 and 9, the indirect effect of attitude on performance mediated by active learning heuristic problem-solving strategies exists with the effect size of 5.18. This effect size is large enough [111]) to conclude that the relationship between students' attitude towards and performance in mathematics word problems exists.

**Table 5. Model summary for performance (post-test).**

| R | R-sq | MSE | F | df1 | df2 | v |
|------|-------|---------|--------|-----|-----|---|
| 0.71 | 0.503 | 165.019 | 306.62 | 2 | 605 | 0 |

**Table 6. Combined model for active learning and attitude.**

|  | Coeff | SE | t | p | LLCI | ULCI |
|---|---|---|---|---|---|---|
| Constant | -43.967 | 4.120 | -10.671 | .000 | -52.059 | -35.876 |
| Attitude | 17.039 | 1.157 | 14.726 | .000 | 14.767 | 19.312 |
| Active Learning | 16.279 | 1.138 | 14.301 | .000 | 14.044 | **18.515** |

Standardized coefficients Attitude .4442, Active Learning .4314.

**Table 7. Direct effect of X on Y.**

| Effect | S.E | t | p | LLCI | ULCI | c'_cs |
|---|---|---|---|---|---|---|
| 17.039 | 1.157 | 14.727 | .000 | 14.767 | 19.312 | .444 |

## 5. Discussions and educational implications

This study explored the relationship between students' attitude towards, and performance in LP mathematics word problems, mediated by active learning heuristic problem-solving (ALHPS) approach. Students' attitude towards mathematics has been a central concern for the last five decades. To date, attitude towards mathematics is still being studied due to the affective role it plays in mathematics education. Studies on attitude towards mathematics have been conducted globally (e.g., [11,13,28,125]). Although there are limitations in assessing students' attitude towards mathematics the ATMI-SF, a reduced scale with fewer subdomains yielded significant results and a better fit to the data. According to [27], students' attitudes towards mathematics may result in students leaving school with a positive emotional disposition and confidence towards mathematics. However, this greatly depends on the instructional methods as defined in different mathematics curricula.

In this study, preliminary analysis revealed that the psychometric properties of the adapted ATLPWTs and the ALHPS approach instruments were found acceptable. The data was collected from 608 students from eight secondary schools in eastern Uganda and central Uganda. The data analysis revealed existence of a direct significant positive relationship between students' performance and their attitude towards learning LP mathematics word problems mediated by active learning heuristic problem solving approach. These findings are of great educational implications to the learning of mathematics in Uganda, and supplements other empirical findings (e.g., [32]). Thus, by boosting students' positive attitude towards mathematics, this may change their performance in mathematics word problems and mathematics generally.

This study indicates that students' attitude enhanced their performance in learning LP concepts. The effectiveness of several approaches aimed at altering students' negative attitude

**Table 8. Indirect effect(s) of X on Y.**

|  | Effect | Boot SE | Boot LLCI | Boot ULCI |
|---|---|---|---|---|
| AL | 5.178 | .641 | 3.968 | 6.454 |

**Table 9. Completely standardized indirect effect(s) of X on Y.**

|  | Effect | Boot SE | Boot LLCI | Boot ULCI |
|---|---|---|---|---|
| AL | 135 | .016 | .104 | .166 |

towards learning mathematics greatly depend not only on students but also on how educators react to and incorporate curriculum changes. The findings concord with other previous empirical studies on the contribution of positive attitude in enhancing students' performance [36,55,126,127]. Seen in this way, the findings are further in agreement with the constructivism theoretical framework suggesting that educators should always review previous concepts and use them to construct new knowledge to enhance secondary school attitude in mathematics. The results of this study are likely to inform educational stakeholders in assessing students' ATLPWTs and provide remediation and interventional strategies aimed at creating a conceptual change in students' attitude towards LP and mathematics generally. This will further act as a lens in improving students' achievement, as indicators of students' confidence, motivation, usefulness, and enjoyment in learning LP word problems and mathematics generally.

These findings show that students generally had negative attitude towards LP word problems. Although some students' ratings were below the neutral attitude (3), they indicated the usefulness of LP in their individual daily lives. The experimental group showed a slightly favorable attitude towards LP word problems after an intervention because the problem-solving heuristic instruction was used during instruction as compared to the students from the comparison group who learned LP conventionally. Some students and teachers revealed that LP concepts are more stimulating, require prior conceptual knowledge and understanding of equations and inequalities, and are not interesting to learn just like other mathematics topics. The explanation provided indicated that some teachers do not adequately apply suitable instructional techniques and learning materials to fully explain the concepts. It was, however, observed that teachers encouraged students to constantly practice model formation from word problems to demystify the negative belief that LP is a hard topic, thereby encouraging them to understand LP and related concepts. However, students' performance especially those from the experimental group improved compared to their counterparts from the comparison group who almost had similar attitude towards LP before and after an intervention. The learning of mathematics should promote students' engagement where learning is by doing through practice and that errors and misconceptions should be considered as part and parcel of learning.

The results of this study are consistent with the theoretical framework [93,128] and conceptual framework [94]. To achieve the purpose of this study, teachers in the experimental group varied tasks to examine students' attitude, problem-solving and critical thinking skills. Both the experimental group and the control group acknowledged the fact that LP is a challenging topic (see Appendix 1 in S1 File), although they highly recognized its significance in constructing models, and optimization in real life. The importance of LP rests in its application and thus teachers were tasked to help learners to develop a positive attitude towards mathematics word tasks, to boost their conceptual understanding, reason insightfully, think logically, critically and, coherently. The teachers' competence in applying instructional strategies helped learners from the experimental group to gain deeper and broader insight, conceptual and procedural understanding, reasoning, and positive attitude. The control group in their conventional instruction still perceived LP as one of the hardest topics. Negative attitude was observed as it was indicated in the most learners' ATMI-SF filled questionnaires. Thus, teachers recognized application of active learning by engaging students with frequent classwork in form of exercises and assignments is paramount. In addition, application of prior conceptual knowledge and understanding may favorably help students to develop a positive attitude and perform better in LP mathematics word tasks. Generally, students from the comparison group seemed not to have adequately developed knowledge of problem solving, logical thinking and reasoning. They did not view the learning of LP from a broader perspective beyond passing national examinations at Uganda Certificate of Education.

The results of this study point to important issues to the educational stakeholders in cultivating an early positive attitude in mathematics which is aimed at investigating specific topics from primary to secondary school mathematics. This may be a potential strategy for applying different heuristic problem-solving approaches and strategies to significantly improve students' attitudes and performance. [129] have stated that epistemological beliefs of learners greatly determine the learning strategies that teachers should apply to stimulate their attitude and performance. In this study, the problem-solving heuristic method supported collaboration and discussions between teachers and amongst students during the learning process. Thus, students from the experimental group worked collaboratively in their small groups. The students helped and guided each other hence boosting their attitude and performance. As noted by [130], the teachers' instructional strategies of considering individual students' differences may consequently change students' attitude, thereby providing both academic and social support.

The findings support the recent Ugandan curriculum reforms in lower secondary school curriculum. By boosting students' attitude towards learning mathematics, their deep and broader conceptual understanding, active engagement, during the learning process, exploring problem-solving strategies, critical thinking, logical reasoning, effective communication, effective utilization of technology, supporting individual students' learning gaps and provision of meaningful assessment practices will be guaranteed. Indeed, when the ALHPS approach was applied to the experimental group, the low performing students greatly gained conceptual understanding and also acquired problem-solving skills. This enhanced students' learning and attitude towards mathematics and solving LP word tasks in particular. Besides, the problem-solving heuristic approach applied to the experimental group boosted students' confidence in answering both routine and non-routine problems. Students' fear in comprehending LP word problems and their negative attitude towards answering LP questions decreased. Students were actively involved in problem-solving. This gradually built students' competence and confidence in learning LP and related concepts which significantly fostered students' positive attitude towards learning LP word problems.

[131] Kilpatrick, Swafford, and Findell (2001) developed five interwoven strands for achieving mathematical proficiency. The five strands are conceptual understanding, procedural fluency, strategic competence, adaptive reasoning, and productive disposition [132]). The main objective is to develop and foster students' abilities in the above five strands. Accordingly, proficiency in mathematics problem-solving entails students' holistic achievement of the three domains: cognitive, affective, and psychomotor. Thus, the methods of teaching mathematics should emphasize and promote learners' acquisition and application of knowledge, beliefs, and skills in solving real-life problems. The authors have argued that "proficiency should enable them to cope with the mathematical challenges of daily life and enable them to continue their study of mathematics in high school and beyond" (p. 116). The above five interrelated strands align with the theoretical and conceptual framework and are inevitable for learning mathematics in the sense that they foster, support, and promote students' identification and acquisition of conceptual knowledge, procedural knowledge, critical thinking, and problem-solving abilities. According to [132], students' effective learning "depends fundamentally on what happens inside the classroom as teachers and learners interact over the curriculum" (p. 8).

## 6. Conclusion and future research directions

The purpose of this research was to explore the relationship between students' attitude towards, and performance in LP mathematics word problems. This study generates and

supports the policy implications linked to the recent education and mathematics curriculum reforms in Uganda. The findings provide preliminary insights into the fundamental concepts and provide an introduction to LP concepts for advanced mathematics. Students' attitudes point to issues related to the demographic variables and latent constructs for learning mathematics. Attitude towards mathematics has both significant direct and indirect effect on students' performance mediated by ALHPS strategies. However, some attitudinal dimensions have only direct effect. Wigfield, and colleagues' expectancy-value model of achievement motivation [93,128] supports this claim. The theory is based on the premise that success on specific tasks and the values inherent in those tasks is positively correlated with performance, and attitude. Thus, the ATMI-SF constructs combining motivation, enjoyment, confidence, and usefulness, and related latent variables are good mediators to explain students' success in learning LP and mathematics generally. The educational stakeholders and experts in mathematics education should embrace suitable learning strategies that cultivate a positive attitude towards learning and consequently performance.

## 6.1 Limitations and future research

As earlier mentioned, studies on attitude towards and performance in mathematics have gradually shifted from general to topic domain-specific studies. Thus, instead of investigating the students' attitude towards mathematics generally, the current research focused on exploring the influence of attitude towards and performance in mathematics word problems and LP in particular. This study adopted a quantitative approach. Despite the conceptual, theoretical, and methodological contributions of this study, several limitations must be considered.

First, to gain more insight, we recommend that future researchers should fill the gap through triangulation. The use of qualitative methods such as interviews and observation may provide more evidence on students' experiences in learning LP word problems, including students' emotional experiences and the general latent behavior. This would help enrich the existing body of knowledge on attitude towards and performance in mathematics. The teachers 'attitude towards the domain specific LP word problems is also a potential area for further investigation aimed at improving the instructional strategies. To achieve this, teachers' content knowledge and pedagogical content knowledge of both in-service and pre-service teachers should be investigated. Also, the teachers' professional development programs should emphasize content knowledge and pedagogical content knowledge. Teachers coming together to share learning experiences and strategies, may help to improve students' attitude and the learning of presumably "difficult topics" including LP and mathematics generally. Indeed, teachers need routine professional development support to successfully implement the stated learning activities. Another potential area for further research is the relationship between students' and teachers' demographic factors mediated by active learning heuristic problem-solving strategies, on students' performance and attitude towards LP and mathematics generally. For this reason, we believe that further comparative studies are needed to better understand the relationship between students' attitude and performance in particular mathematics units (topics). A better understanding of these influences is crucial to design actions that promote students' academic achievement.

## Supporting information

**S1 File.**
(DOCX)

## Acknowledgments

This research is part of the Ph.D. Thesis that investigated the effect of the heuristic problem-solving approach on students' achievement and attitude towards LP in Ugandan secondary schools. We appreciate useful information provided by the students and teachers in the study sample, which helped us, write this research article. The views expressed herein are those of the authors and not necessarily those of our funders. This is because our funders were not involved in identifying the suitable study design, methods of data collection and analysis, publication decision, or manuscript preparation.

## Author Contributions

**Conceptualization:** Robert Wakhata.

**Data curation:** Robert Wakhata.

**Formal analysis:** Robert Wakhata.

**Funding acquisition:** Robert Wakhata.

**Investigation:** Robert Wakhata.

**Methodology:** Robert Wakhata.

**Project administration:** Robert Wakhata.

**Resources:** Robert Wakhata.

**Software:** Robert Wakhata.

**Supervision:** Sudi Balimuttajjo, Védaste Mutarutinya.

**Validation:** Robert Wakhata, Sudi Balimuttajjo, Védaste Mutarutinya.

**Visualization:** Robert Wakhata, Sudi Balimuttajjo, Védaste Mutarutinya.

**Writing – original draft:** Robert Wakhata, Sudi Balimuttajjo.

**Writing – review & editing:** Robert Wakhata, Sudi Balimuttajjo, Védaste Mutarutinya.

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
