## [Decision Letter · Decision Letter 0]

27 Feb 2023

PONE-D-22-31844Relationship between Students’ Attitude towards, and Performance in Mathematics Word ProblemsPLOS ONE

Dear Dr. Wakhata,

Thank you for submitting your manuscript to PLOS ONE. After careful consideration, we feel that it has merit but does not fully meet PLOS ONE’s publication criteria as it currently stands. Therefore, we invite you to submit a revised version of the manuscript that addresses the points raised during the review process.

Peer review of your manuscript is now complete and, in light of the reports, and my own assessment as an academic editor, I inform you that your manuscript needs major revisions. Please find the reviewers' reports at the end of this email. As you can see, both reviewers shared their questions and concerns about the study.  Please pay more attention to reviewer 2’s comments when you are revising and improving your paper.

We look forward to receiving your revised manuscript.

Kind regards,

Daner Sun

Academic Editor

PLOS ONE

Journal Requirements:

2. Please provide additional details regarding participant consent. In the Methods section, please ensure that you have specified (1) whether consent was informed and (2) what type you obtained (for instance, written or verbal). If your study included minors, state whether you obtained consent from parents or guardians. If the need for consent was waived by the ethics committee, please include this information

3. Please change "female” or "male" to "woman” or "man" as appropriate, when used as a noun (see for instance https://apastyle.apa.org/style-grammar-guidelines/bias-free-language/gender)

4. We note that you have stated that you will provide repository information for your data at acceptance. Should your manuscript be accepted for publication, we will hold it until you provide the relevant accession numbers or DOIs necessary to access your data. If you wish to make changes to your Data Availability statement, please describe these changes in your cover letter and we will update your Data Availability statement to reflect the information you provide

Reviewers' comments:

Reviewer's Responses to Questions

**Comments to the Author**

1. Is the manuscript technically sound, and do the data support the conclusions?

Reviewer #1: Yes

Reviewer #2: Partly

2. Has the statistical analysis been performed appropriately and rigorously? 

Reviewer #1: Yes

Reviewer #2: No

3. Have the authors made all data underlying the findings in their manuscript fully available?

Reviewer #1: Yes

Reviewer #2: No

4. Is the manuscript presented in an intelligible fashion and written in standard English?

Reviewer #1: Yes

Reviewer #2: Yes

5. Review Comments to the Author

Reviewer #1: The study presents the results of original research. Experiments, statistics, and other analyses are performed to a good technical standard and are described in sufficient detail. Conclusions are presented in an appropriate fashion and are supported by the data. The article is presented in an intelligible fashion and is written in standard English.

Still, there are some issues for the authors' consideration and revisions.

Para 1.1 Please revise the citation format in the first sentence.

Para 3.2 Of the 608 students, 291 (47.86%) were males and 492 (57.8%) were females with a mean age of 18.36 (S.D = 0.94) years. 291 plus 492 does not equal 608. Please double check.

For the Data Analysis part, please present the tables according the APA style. For example, it should be three digits after decimal point rather than four digits. The data analysis part needs to be enhanced.

Reviewer #2: This research seems to be interesting, but the author did not introduce it following rigorous academic norms. The theoretical basis, methodology, analysis and discussion of the results have not been fully delivered. In the literature review section, many ideas are vague and need to be expanded. In the methodology section, does the research design effectively respond to the purpose of the research, and does each subsection follow a logical sequence? The discussion section lacks systematic analysis, as well as critical discussion. In the Results section, there is no response to the purpose of the study. The following are some specific comments including textual and logical issues:

P3, 1.1 Verschaffel et al., 2010),

P5, 1.3 there is no “in the middle of the sentence.

P2 and P3, there are many references supporting one statement, which is a very general conclusion, for example, "some empirical findings report students’ attitudes towards specific units or topics in mathematics to enhance the learning of that specific content and mathematics”, and “Yet, some empirical studies have shown existence of significant

relationships between students’ attitude and performance in mathematics”. Do they say the same thing or have the same effect? And could you kindly clarify this statement and make those references indicating the specific results?

P3, “it appears that multiple factors ranging from students’ demographic factors to teachers’ classroom instructional practices influence students’ attitude towards learning mathematics" What multiple factors? I would suggest authors indicate this clearly.

P4, “Despite numerous difficulties encountered by students in algebraic inequalities (e.g., Fernández & Molina, 2017; Molina et al., 2017;Bazzini & Tsamir, 2004; Tsamir & Almog, 2001; Tsamir & Bazzini, 2004, 2006; Tsamir & Tirosh, 2006), a combination of methods (strategies) rather than one specific method can be applied to overcome specific students’ learning challenges.” These sentences seem not to be connected with the prior sentences. Please re-organize this part so that it could be aligned with this section.

Section 1.2 is too simple, authors should expand some points, for example, “Previous empirical studies have revealed that LP and/or related concepts are not only difficult for learners but also challenging to teach (Awofala, 2014; Goulet-Lyle et al., 2020; Kenney et al., 2020; Verschaffel et al., 2020a, 2020b).” What difficulties did the students encounter, cognitive or non-cognitive? “Different factors account for learners’ challenges in mathematics word problems?” What are these different factors? Is there existing research on attitudes towards Mathematics Word Problems? What kind of attitudes students held?

P6, not sure how these two theoretical frameworks connected. Could you clarify how "expectancy-value model theory within the constructivism paradigm” works?

P7, “Active Learning Heuristic Problem Solving Strategies" is here for the first time. More information should be provided in the literature review about these strategies.

Methodology, authors should make sure the correct methodology, “A quantitative survey research design” or “A quasi-experimental pre-test, post-test, non-equivalent, non-randomized control group study design". Please check the methodology and method. They are different.

P9, Treatment group vs. Experimental group, please use the same term throughout the context.

P9, “The learning in the comparison group was purely conventional, and teachers did not follow proper and organized strategies as was the case with the treatment group.” It seems that the treatment group has been involved in a more scientific, systematic, and better teaching method than the comparison group, so it is certain that the treatment group will achieve better results. In this case, the conclusion is obvious.

“Empirical studies (e.g., Abdul et al., 2010; Sari et al., 2012) have found that the stated materials enhance students’ conceptual understanding and critical thinking abilities. These strategies were further integrated in problem solving strategies (Polya, 2014) by ensuring that students understand the LP word problem, devise a plan, adequately carry out the plan and finally look back to verify solution sketches (see Appendix 3 for PS01, PS02, PS03, PS04). To ensure that students minimize errors and misconceptions, the learning of LP was further integrated with Newman Error Analysis (NEA) model (Mushlihah, 2018)." Those statements should be put in the literature review.

“The instrument was designed by the researcher and validated by experts in mathematics education as an error analysis tool to provide teachers with a framework to consider the underlying reasons why students answer mathematics word problems incorrectly. The teachers emphasized question reading and decoding, comprehension, transformation, process skills and encoding (see Appendix 3 for NEA01, NEA02, NEA03, NEA04 and NEA05).” You started to say new points. This paragraph contains two many ideas: strategies, instruments, coding.

P10. “the post attitude questionnaire and achievement test were administered to both groups, and analysis was done by comparing and contrasting students’ feedback on attitudinal constructs and the post-test scores." Not sure what the achievement test is. You should clearly state how many instruments you apply and how many hours implement the treatment group. Who implemented it in the treatment group and comparison group?

For table 4 and table 5, there is not enough explanation, what statistical method is used, and how to explain the data?

Again, the description of table8,9,10 is also very simple and unclear.

The main problem in the discussion part is that there is a lack of comparison with previous literature, and critical discussion, as with the total number of literature, the author needs to make a more detailed comparison with the relevant literature instead of citing a lot of literature.

The conclusion section does not respond well to the purpose of the study.

6. PLOS authors have the option to publish the peer review history of their article (what does this mean?). If published, this will include your full peer review and any attached files.

Reviewer #1: No

Reviewer #2: No

---

## [Author Response · Author response to Decision Letter 0]

27 Sep 2023

Revision of the manuscript titled “Relationship between Students’ Attitude towards, and Performance in Mathematics Word Problems” 

Re: Response to Reviewer’s Comments on the Manuscript PONE-D-22-31844

From: "PLOS ONE" plosone@plos.org

We appreciate you for your precious time that was dedicated in reviewing our manuscript and providing valuable comments. Thank you for giving us the opportunity to submit a revised version of our manuscript titled “Relationship between Students’ Attitude towards, and Performance in Mathematics Word Problems”. These valuable and insightful comments have led to great improvements in the current version. We have been able to incorporate changes to reflect most of the suggestions provided by the esteemed reviewers. We have collectively and carefully considered your comments aimed at improving the quality of our paper and we have tried to address each of them. We hope that our manuscript after careful revision especially with your guidance will meet your standards. However, should there be more points not adequately addressed, we kindly welcome further constructive input. 

Addressed below are point-by-point responses. All modifications in the manuscript have been highlighted in yellow. 

Journal Requirements:

1. Please ensure that your manuscript meets PLOS ONE's style requirements, including those for file naming. Thank you, this has been revised although I did not get a word document.

2. Please provide additional details regarding participant consent. In the Methods section, please ensure that you have specified (1) whether consent was informed and (2) what type you obtained (for instance, written or verbal). If your study included minors, state whether you obtained consent from parents or guardians. If the need for consent was waived by the ethics committee, please include this information. Details of consent have been included as proposed.

3. Please change "female” or "male" to "woman” or "man" as appropriate, when used as a noun (see for instance https://apastyle.apa.org/style-grammar-guidelines/bias-free-language/gender). This was not changed as suggested because the data was for students under 20 years and are locally addressed as females and males and not women and males.

4. We note that you have stated that you will provide repository information for your data at acceptance. Should your manuscript be accepted for publication, we will hold it until you provide the relevant accession numbers or DOIs necessary to access your data. If you wish to make changes to your Data Availability statement, please describe these changes in your cover letter and we will update your Data Availability statement to reflect the information you provide. This has now been included.

Reviewers' comments:

Reviewer's Responses to Questions

Comments to the Author

Reviewer #1

Para 1.1 Please revise the citation format in the first sentence.

Para 3.2 Of the 608 students, 291 (47.86%) were males and 492 (57.8%) were females with a mean age of 18.36 (S.D = 0.94) years. 291 plus 492 does not equal 608. Please double check. This has been addressed

For the Data Analysis part, please present the tables according the APA style. For example, it should be three digits after decimal point rather than four digits. The data analysis part needs to be enhanced. The tables and analysis section has been enhanced

Reviewer #2: 

This research seems to be interesting, but the author did not introduce it following rigorous academic norms. The theoretical basis, methodology, analysis and discussion of the results have not been fully delivered. In the literature review section, many ideas are vague and need to be expanded. In the methodology section, does the research design effectively respond to the purpose of the research, and does each subsection follow a logical sequence? The discussion section lacks systematic analysis, as well as critical discussion. In the Results section, there is no response to the purpose of the study. The following are some specific comments including textual and logical issues:

P3, 1.1 Verschaffel et al., (2010), has been corrected

P5, 1.3 there is no “in the middle of the sentence. Was corrected

P2 and P3, there are many references supporting one statement, which is a very general conclusion, for example, "some empirical findings report students’ attitudes towards specific units or topics in mathematics to enhance the learning of that specific content and mathematics”, and “Yet, some empirical studies have shown existence of significant

relationships between students’ attitude and performance in mathematics”. Do they say the same thing or have the same effect? And could you kindly clarify this statement and make those references indicating the specific results? Was clarified and additional literature included.

P3, “it appears that multiple factors ranging from students’ demographic factors to teachers’ classroom instructional practices influence students’ attitude towards learning mathematics" What multiple factors? I would suggest authors indicate this clearly. These factors were cited.

P4, “Despite numerous difficulties encountered by students in algebraic inequalities (e.g., Fernández & Molina, 2017; Molina et al., 2017;Bazzini & Tsamir, 2004; Tsamir & Almog, 2001; Tsamir & Bazzini, 2004, 2006; Tsamir & Tirosh, 2006), a combination of methods (strategies) rather than one specific method can be applied to overcome specific students’ learning challenges.” These sentences seem not to be connected with the prior sentences. Please re-organize this part so that it could be aligned with this section. This has been improved.

Section 1.2 is too simple, authors should expand some points, for example, “Previous empirical studies have revealed that LP and/or related concepts are not only difficult for learners but also challenging to teach (Awofala, 2014; Goulet-Lyle et al., 2020; Kenney et al., 2020; Verschaffel et al., 2020a, 2020b).” What difficulties did the students encounter, cognitive or non-cognitive? “Different factors account for learners’ challenges in mathematics word problems?” What are these different factors? Is there existing research on attitudes towards Mathematics Word Problems? What kind of attitudes students held? Explanations and additional literature included

P6, not sure how these two theoretical frameworks connected. Could you clarify how "expectancy-value model theory within the constructivism paradigm” works? The two theories have been linked

P7, “Active Learning Heuristic Problem Solving Strategies" is here for the first time. More information should be provided in the literature review about these strategies. The manuscript was written to publish just one of the objectives of this major study. This has been explained.

Methodology, authors should make sure the correct methodology, “A quantitative survey research design” or “A quasi-experimental pre-test, post-test, non-equivalent, non-randomized control group study design". Please check the methodology and method. They are different. A quantitative was adopted. This has been clarified.

P9, Treatment group vs. Experimental group, please use the same term throughout the context. Thank you, this has been used consistently.

P9, “The learning in the comparison group was purely conventional, and teachers did not follow proper and organized strategies as was the case with the treatment group.” It seems that the treatment group has been involved in a more scientific, systematic, and better teaching method than the comparison group, so it is certain that the treatment group will achieve better results. In this case, the conclusion is obvious. No, it is not obvious but depends on the findings and the decision on whether or not the Null hypothesis is rejected or accepted. There are cases when an intervention does not yield significant results.

“Empirical studies (e.g., Abdul et al., 2010; Sari et al., 2012) have found that the stated materials enhance students’ conceptual understanding and critical thinking abilities. These strategies were further integrated in problem solving strategies (Polya, 2014) by ensuring that students understand the LP word problem, devise a plan, adequately carry out the plan and finally look back to verify solution sketches (see Appendix 3 for PS01, PS02, PS03, PS04). To ensure that students minimize errors and misconceptions, the learning of LP was further integrated with Newman Error Analysis (NEA) model (Mushlihah, 2018)." Those statements should be put in the literature review. This has been re-arranged accordingly

“The instrument was designed by the researcher and validated by experts in mathematics education as an error analysis tool to provide teachers with a framework to consider the underlying reasons why students answer mathematics word problems incorrectly. The teachers emphasized question reading and decoding, comprehension, transformation, process skills and encoding (see Appendix 3 for NEA01, NEA02, NEA03, NEA04 and NEA05).” You started to say new points. This paragraph contains two many ideas: strategies, instruments, coding. This has been improved.

P10. “the post attitude questionnaire and achievement test were administered to both groups, and analysis was done by comparing and contrasting students’ feedback on attitudinal constructs and the post-test scores." Not sure what the achievement test is. You should clearly state how many instruments you apply and how many hours implement the treatment group. Who implemented it in the treatment group and comparison group? This has been explained in methodology section

For table 4 and table 5, there is not enough explanation, what statistical method is used, and how to explain the data? Additional explanations have been provided

Again, the description of table8,9,10 is also very simple and unclear. I have tried to added few lines for clarity

The main problem in the discussion part is that there is a lack of comparison with previous literature, and critical discussion, as with the total number of literature, the author needs to make a more detailed comparison with the relevant literature instead of citing a lot of literature. We have tried to discuss and provide relevant additional literature.

The conclusion section does not respond well to the purpose of the study. This has been improved.

Thank you

Robert Wakhata

Corresponding Author

---

## [Editor Report · Decision Letter 1]

28 Sep 2023

Relationship between Students’ Attitude towards, and Performance in Mathematics Word Problems

PONE-D-22-31844R1

Dear Dr. Wakhata,

We’re pleased to inform you that your manuscript has been judged scientifically suitable for publication and will be formally accepted for publication once it meets all outstanding technical requirements.

Kind regards,

Daner Sun

Academic Editor

PLOS ONE

Additional Editor Comments (optional):

The authors have well addressed the reviewers' comments.
---

## [Editor Report · Acceptance letter]

13 Nov 2023

PONE-D-22-31844R1 

Relationship between Students’ Attitude towards, and Performance in Mathematics Word Problems 

Dear Dr. Wakhata:

I'm pleased to inform you that your manuscript has been deemed suitable for publication in PLOS ONE. Congratulations! Your manuscript is now with our production department. 

Kind regards, 

on behalf of

Dr. Daner Sun 

Academic Editor

PLOS ONE